# The Effects of Wearing Helmets on Reaction of Motorcycle Riders

**Dengchuan Cai** [1], **Yu-Hsuan Chen** [2] **and Chih-Jen Lee** [1,*] 

1   Graduate School of Design, National Yunlin University of Science and Technology, 123 University Rd. Sec. 3, Douliu 640, Taiwan; caidc@yuntech.edu.tw

2   Department of Industrial Design, National Yunlin University of Science and Technology, 123 University Rd. Sec. 3, Douliu 640, Taiwan; m10731009@gmail.com

*   Correspondence: g9430809@yuntech.edu.tw; Tel.: +886-931-528811

**Abstract:** In Taiwan, motorcycles are the most commonly used means of transportation and also have the highest accident rate. Because motorcycles are less stable and provide less protection than cars, motorcycle riders are vulnerable in traffic accidents. Furthermore, head trauma is often fatal, causing a great loss to society. Although helmets provide protection to the head, they also affect the visual field of motorcycle riders. However, the literature mostly focuses on the protective effect of helmets after a collision and rarely considers the influence of helmets prior to collisions. In the study design, participants wore three different types of helmet and watched a pre-recorded video of an actual street with pre-placed warning triangles at a speed of 60 km/h. Participants were asked to press a button when they saw a warning triangle. The time between participants seeing the warning triangle and arriving at the warning triangle was calculated. This time is referred to as the "early reaction time." The number of missed presses and false presses was also recorded. The results of the study show that: (1) Of the three types of helmet, wearing half helmets produced the longest early reaction times, followed by 3/4 helmets, with full face helmets with the shortest early reaction times. (2) Early reaction times when wearing a half helmet were the same as early reaction times when not wearing a helmet. (3) The results for the total number of missed and false presses when wearing the three types of helmet were the same as for the results of the early reaction time analysis. (4) Sex and age had no effect on early reaction times. The experimental results can be used as a reference for helmet design and academic research.

**Keywords:** helmet; motorcycle rider; early reaction time



## 1. Introduction

According to the National Travel Survey carried out by the Ministry of Transportation and Communications [1], in 2016, the most commonly used mode of transportation in Taiwan was the motorcycle (45.9%). According to data released by the National Police Agency [2], there were 238,780 deaths and injuries involving motorcycle riders in 2014, accounting for 57.53% of traffic accident deaths and injuries. In addition, the percentage of traffic accident deaths and injuries involving motorcycle rides exceeded 50% for three consecutive years starting in 2012. Based on the above data, motorcycles are the most commonly used means of transportation and also have the highest accident rate. Although motorcycles have the advantages of being economical and convenient, because motorcycles are less stable and provide less protection than cars, motorcycle riders are vulnerable in traffic accidents. Furthermore, head trauma is often fatal, causing great loss to society. After Taiwan enforced the mandatory use of helmets for motorcycle riders and passengers on 1 June 1997 [3], the helmet usage rate has reached 80%. In addition, the criteria for approved helmets and the manner in which helmets are to be worn are also clearly stipulated [4]. As a result, helmets have become an essential safety item when riding motorcycles in Taiwan.

In Taiwan, helmets must pass the National Standards of the Republic of China (CNS) and the Product Safety Mark [5], which are divided into two types of helmet: normal (suitable for non-racing motorcycles under 125cc) and strengthened (suitable for non-racing motorcycles over 125cc). In addition, helmets approved by the United States Department of Transportation (DOT) [6], Economic Commission of Europe (ECE) [7], SNELL Foundation 2020 Helmet Standard For Use in Motorcycling (SNELL M2020) [8], and Japanese Industrial Standards (JIS) [9] can also be sold. The DOT adopts independent certification and conducts random testing after the market launch, and is the most common helmet certification in the world. SNELL's certification process is more stringent than DOT, and helmets that pass the DOT standards may not pass SNELL testing. In addition, SNELL purchases helmets on the market with SNELL labels for random retesting [10].

Although helmets provide protection to the head, they also affect the visual field of motorcycle riders. Currently, helmets available on the market can be divided into half helmets, 3/4 helmets, and full face helmets. Different types of helmets will produce different obstructions to the visual field of riders, and this visual field is closely related to riders' reaction times.

Robers [11] described the sum of the time for a driver to perceive, assess, judge, and react to road conditions as the response time. This response time is approximately 2–4 s. Nicholas and Lester [12] found that reaction times are an important factor in determining braking distance and are related to the stopping sight distance. When the reaction time is too slow, the driver is unable to brake in time, causing an accident. In other words, when drivers react more quickly, they have more time to make an appropriate judgment (such as braking or taking evasive action).

Other factors affecting riders' early reaction times include the rider's mental state and age, road pavement, traffic intensity, and weather conditions.

Research on driving fatigue generally argues that the length of time spent driving is the most relevant factor. The fatigue effect begins to occur when driving continuously for more than eight hours [13,14]. In addition, according to medical data, the body begins to experience physical fatigue after one hour of monotonous and repetitive driving regardless of whether the individual is aware of it [15]. To avoid the occurrence of these factors that may affect the results of the experiment, we ensured that participants were in good mental and physical condition at the time of the experiment, and the total length of the video was limited to ten minutes.

Age is considered to be one of the influencing factors in previous studies on the physiological condition of drivers. Older drivers are less able to perceive potential road hazards due to deteriorating physiological function (e.g., concentration, effective visual field, physical strength, and reflexes) [16]. However, Borowsky et al. [17] analyzed the effect of age and driving experience on the ability to detect hazards, finding that older and more experienced drivers are more sensitive to hazardous situations than younger drivers. In addition, statistics have shown that older drivers (aged 60 and over) account for 15.9% of fatalities and injuries, compared to 43.2% involving younger drivers (aged 18–29) [18].

Currently, most road pavements in Taiwan use asphalt concrete pavement and portland cement concrete pavement. The former is constructed in accordance with the "Standard Specifications for Highway Construction Section 02742" [19] and AASHTO [20]. The latter is constructed in accordance with the "Standard Specifications for Highway Construction Section 02751" [19] and ASTM [21]. Therefore, road pavements have fatigue resistance (the ability to prevent the pavement breaking up due to the bending effect of repeated vehicle loads) and skid resistance (resistance to sliding when the brakes are applied), reducing the risk of riders sliding and providing safe road performance for road users.

Past studies have shown that the accident frequency of each road segment is mainly influenced by factors related to annual average daily traffic, road geometry, and weather conditions [22–25]. However, there is strong evidence showing that human factors are the most important influencing factor in road traffic accidents [26–31]. Distraction and inattention are the two most important human factors in road traffic accidents.

Based on the above, the participants in this study were young riders who have a higher ratio of accidents. Watching a pre-recorded video excluded factors such as road pavement, traffic intensity, and weather conditions, allowing us to focus on the field of vision and early reaction time when wearing helmets.

However, the literature mostly focuses on the protective effect of helmets after a collision and rarely considers the influence of helmets prior to collisions. Therefore, the present study examined riders' early reaction times, false presses, missed presses, sex, and age for three types of helmets, analyzing possible influence factors as a starting point for improving helmet design.

## 2. Methods

### 2.1. Participants

The present study recruited 52 adult participants, with 26 participants of each sex. The age of participants was between 18–35 years old, with an average age of 23.24 years old (SD = 3.54 years old). All participants had motorcycle licenses and had 1.0 vision of better after correction. The participants had good mental and physical condition at the time of the experiment and did not stay up late or overwork prior to participating in the experiment.

### 2.2. Experimental Variables

Independent variables: The two independent variables in the study were as follows:

1. Sex: male, female
2. Types of helmet: full face helmets, 3/4 helmets, half helmets, as shown in Figure 1.

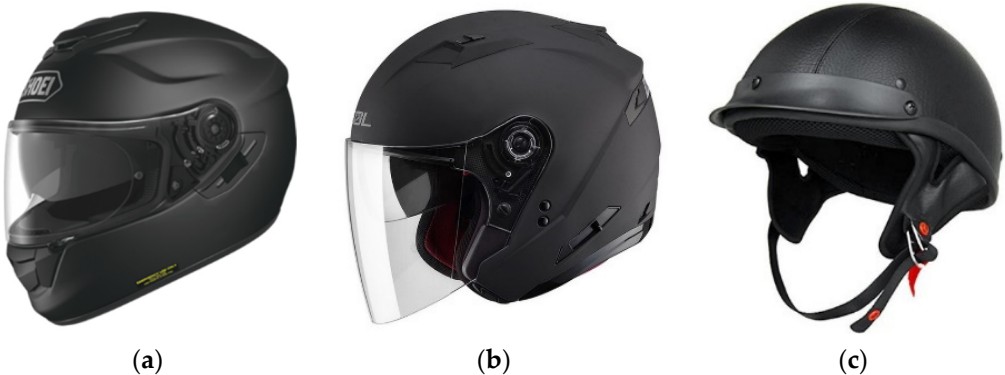

| (a) | (b) | (c) |

**Figure 1.** The Three Types of Helmet Used in the Experiment. (**a**) Full face helmet, had the narrowest visual field; (**b**) 3/4 helmet, had the next narrowest visual field; (**c**) Half helmet, had the largest visual field.

Dependent variables: The three dependent variables in the study were as follows:

1. Early reaction time: The time between the participant seeing the warning triangle and arriving at the warning triangle. The video was filmed at a speed of 60 km/h.
2. Number of false presses: The number of times throughout the experiment that the participant pressed the button when there was no warning triangle in the video.
3. Number of missed presses: The number of times throughout the experiment that the participant did not press the button when a warning triangle appeared in the video.

### 2.3. Apparatus and Procedures

1. ① Preparation before recording video: The present study first randomly placed a total of 10 real warning triangles measuring 35 cm × 39 cm on the left or the right side of the planned route (as shown in Figure 2). ② Video recording: A GoPro 7 camera was used to record the video from a car traveling at a speed of 60 km/h (as shown in Figure 3). The total distance was 10 km. The total duration of the video was 10 min.

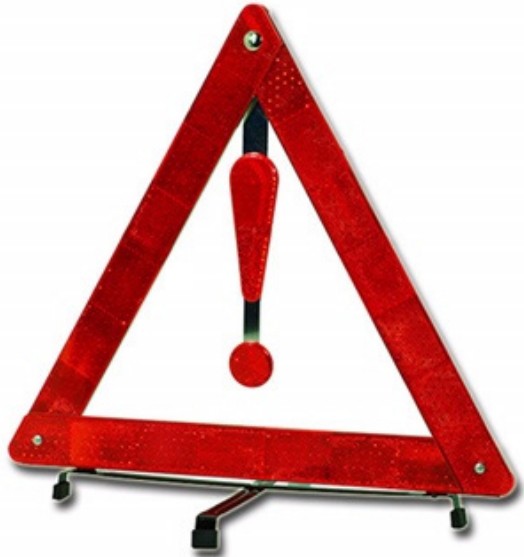

**Figure 2.** Warning Triangle.

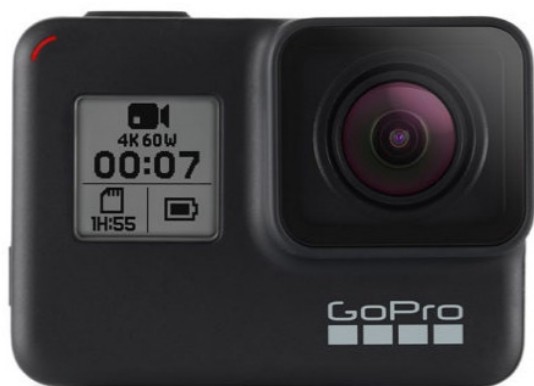

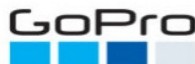

**Figure 3.** GoPro 7 Camera.

2.   ① Participants' visual acuity was measured before the experiment. Participants with corrected visual acuity of better than 1.0 were allowed to perform the experiment. ② Participants sat in front of a table 150 cm away from the screen. The height of the chair was adjusted so that participants' eyes were at the same height as the center of the screen. Participants were informed that the experiment was a video viewing experiment to simulate reactions when riding a motorcycle. Warning triangles appeared on both sides of the road in the video. Participants were asked to press a button when they saw a warning triangle to confirm that they had seen the warning triangle (as shown in Figure 4). ③ A computer program was used to control the playback of the video. When participants pressed the button, the time and the location of the press were recorded. ④ At the end of the experiment, the early reaction time (as shown in Figure 5), number of false presses, and number of missed presses of each participant were recorded.

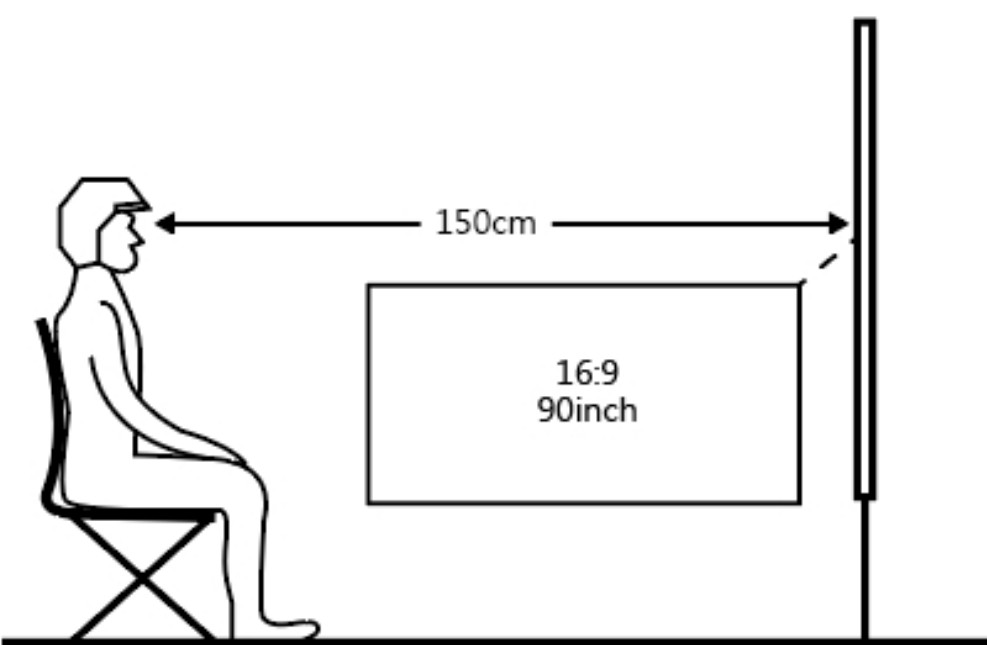

**Figure 4.** Helmet and Reaction Experiment.

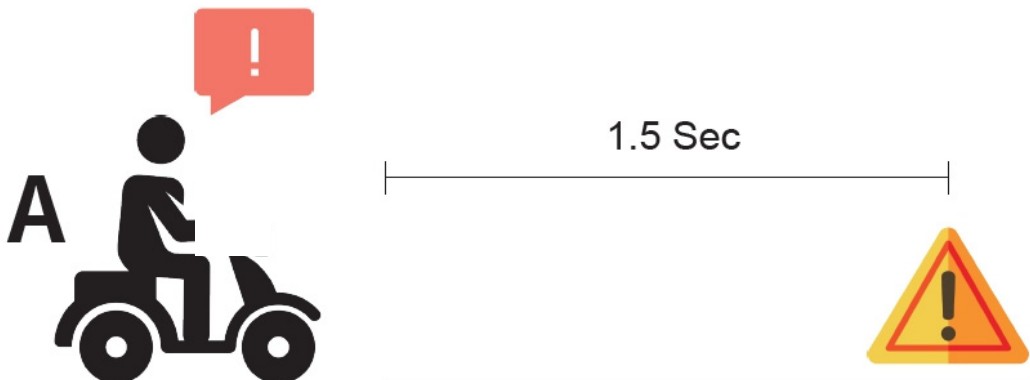

**Figure 5.** Participants' early reaction times.

*2.4. Experimental Design*

The present study conducted two experiments. Experiment 1 was effect of wearing the three types of helmet on riders' reactions. Experiment 2 was the effect of not wearing a helmet on riders' reactions.

Experiment 1 used a within-subject design. There was a total of 40 participants. Participants were required to wear the three types of helmet mentioned above for the early reaction time and missed presses experiment.

Experiment 2 was the control group for experiment 1. There were 12 participants whose early reaction time and missed presses experiment were tested when not wearing helmets. The results were then compared with experiment 1.

The analysis of the two experiments was conducted by a statistical analysis t-test. The t-test was proposed by William Sealy Gosset (pseudonym "Student") in 1908 [32]. When conducting research, the t-test is the most commonly used statistical method to determine whether there is a significant difference between two sets of data [33]. The experimental group and control group can be considered to be two independent samples for testing the hypothesis using the sample means (two independent samples test) [34].

## 3. Results and Discussions

### 3.1. Early Reaction Time

The early reaction times for the three types of helmets are shown in Table 1. Half helmets had the longest early reaction time, with the object detected on average 1.622 s in advance. 3/4 helmets had the next longest early reaction time, with the object detected on average 1.395 s in advance. Full helmets had the shortest early reaction time, with the object detected on average 1.115 s in advance.

**Table 1.** Early reaction times for each type of helmet (unit: seconds).

| Type | N | Min | Max | Mean | SD |
|---|---|---|---|---|---|
| Full face helmet | 479 | 0.01 | 3.79 | 1.115 | 0.576 |
| 3/4 helmet | 511 | 0.17 | 3.80 | 1.395 | 0.554 |
| Half helmet | 517 | 0.24 | 3.97 | 1.622 | 0.592 |

According to Triggs and Harris [35], the 85th percentile distribution of driver braking time is between 1.26 and 3 s. In the present study, only the early reaction time of half helmets was within the 85th percentile distribution of braking time as measured by Triggs and Harris.

Paired t-tests were used to test the early reaction times of the three types of helmet. The test results are shown in Table 2. The early reaction time wearing full face helmets was shorter than when wearing 3/4 helmets (t = 21.54, $p < 0.01$) and half helmets (t = 28.38, $p < 0.01$). The early reaction time when wearing 3/4 helmets was shorter than when wearing half helmets (t = 18.54, $p < 0.01$).

**Table 2.** Paired sample t-tests for early reaction times when wearing different types of helmet (unit: seconds).

| Type | N | Mean | SD | t | P |
|---|---|---|---|---|---|
| Full face helmet | 359 | 0.961 | 0.485 | 21.54 | 0.000 |
| 3/4 helmet | 391 | 1.339 | 0.533 | | |
| 3/4 helmet | 391 | 1.339 | 0.533 | 18.54 | 0.000 |
| Half helmet | 397 | 1.642 | 0.596 | | |
| Full face helmet | 359 | 0.961 | 0.485 | 28.38 | 0.000 |
| Half helmet | 397 | 1.642 | 0.596 | | |

Of the three types of helmet, full face helmets had the shortest early reaction time, 3/4 helmets had the next shortest early reaction time, and half helmets had the longest early reaction time. Therefore, wearing full face helmets shortens riders' early reaction time in case of an incident.

### 3.2. Analysis of Missed Presses

Paired t-tests were used to test the number of missed presses for full face helmets, 3/4 helmets, and half helmets. The test results are shown in Table 3. The results show that wearing full face helmets produced significantly more missed presses than wearing 3/4 helmets (t = 4.051, $p < 0.01$) and half helmets (t = 2.243, $p = 0.031$). In addition, the number of missed presses when wearing 3/4 helmets was significantly greater than when wearing half helmets (t = 5.46, $p < 0.01$). This result was consistent with the early reaction time statistical results.

**Table 3.** Paired sample t-tests for missed presses when wearing different types of helmet (unit: times).

| Type | N | Mean | SD | t | P |
|---|---|---|---|---|---|
| Full face helmet | 40 | 1.00 | 1.155 | 4.051 | 0.000 |
| 3/4 helmet | 40 | 0.28 | 0.751 | | |
| 3/4 helmet | 40 | 0.28 | 0.751 | 2.243 | 0.031 |
| Half helmet | 40 | 0.08 | 0.35 | | |
| Full face helmet | 40 | 1.00 | 1.155 | 5.460 | 0.000 |
| Half helmet | 40 | 0.08 | 0.35 | | |

### 3.3. Analysis of False Presses

Paired t-tests were used for statistical analysis of the number of false presses for full face helmets, 3/4 helmets, and half helmets. The test results are shown in Table 4. The results show that the number of false presses was higher when wearing full face helmets than when wearing 3/4 helmets (t = 6.271, $p < 0.01$) and half helmes (t = 6.993, $p < 0.01$), and the number of false presses when wearing 3/4 helmets was higher than when wearing half helmets (t = 2.912, $p = 0.06$). This result was consistent with the early reaction time and missed presses statistical results.

**Table 4.** Paired sample t-tests for false presses when wearing different types of helmet (unit: times).

| Type | N | Mean | SD | t | P |
|---|---|---|---|---|---|
| Full face helmet | 40 | 1.15 | 1.001 | 6.271 | 0.000 |
| 3/4 helmet | 40 | 0.28 | 0.506 | | |
| 3/4 helmet | 40 | 0.28 | 0.506 | 2.912 | 0.006 |
| Half helmet | 40 | 0.03 | 0.158 | | |
| Full face helmet | 40 | 1.15 | 1.001 | 6.993 | 0.000 |
| Half helmet | 40 | 0.03 | 0.158 | | |

### 3.4. Early Reaction Times by Sex

To explore the influence of sex on early reaction time, the early reaction time when wearing three types of helmet was examined with sex as a stratification variable. The results showed no significant difference for full face helmets (t = 1.481, $p = 0.140$), 3/4 helmets (t = 1.117, $p = 0.261$). and half helmets (t = 1.615, $p = 0.104$), as shown in Table 5.

**Table 5.** Independent t-tests for sex differences in early reaction times (unit: seconds).

| Type | Sex | N | Mean | SD | t | P |
|---|---|---|---|---|---|---|
| Full face helmet | Male | 20 | 0.99 | 0.50 | −1.481 | 0.140 |
| | Female | 20 | 0.92 | 0.47 | | |
| 3/4 helmet | Male | 20 | 1.37 | 0.58 | −1.117 | 0.261 |
| | Female | 20 | 1.31 | 0.49 | | |
| Half helmet | Male | 20 | 1.68 | 0.66 | −1.615 | 0.104 |
| | Female | 20 | 1.58 | 0.53 | | |

### 3.5. Early Reaction Times by Age

The present study divided participants into two age groups: 18–25 years old and 26–35 years old, examining early reaction times when wearing the three types of helmet with age as an independent variable. The results showed no significant difference in early reaction times when wearing full face helmets (t = 1.257, $p = 0.210$), 3/4 helmets (t = 1.581, $p = 0.115$). and half helmets (t = 1.013, $p = 0.311$), as shown in Table 6. This finding indicates that there was no difference in early reaction times between the age groups in the present study, perhaps because the difference in age between the groups was too small.

**Table 6.** Independent t-tests for age differences in early reaction times (unit: seconds).

| Type | Age | N | Mean | SD | t | P |
|---|---|---|---|---|---|---|
| Full face helmet | 18 to 25 | 288 | 0.976 | 0.50 | −1.257 | 0.210 |
|  | 26 to 35 | 71 | 0.894 | 0.40 |  |  |
| 3/4 helmet | 18 to 25 | 311 | 1.360 | 0.55 | −1.581 | 0.115 |
|  | 26 to 35 | 80 | 1.253 | 0.45 |  |  |
| Half helmet | 18 to 25 | 317 | 1.649 | 0.62 | −1.013 | 0.311 |
|  | 26 to 35 | 80 | 1.573 | 0.49 |  |  |

*3.6. Analysis Early Reaction Times with and without Helmet*

To investigate the effects of wearing a helmet on early reaction time, the present study conducted independent t-tests to compare the early reaction time when wearing the three types of helmet and when not wearing a helmet. The results are shown in Table 7. The table shows that the early reaction times when wearing full face helmets (t = 10.06, $p < 0.01$) and 3/4 helmets (t = 4.12, $p < 0.01$) were significantly shorter than the early reaction times when not wearing a helmet. In contrast, there was no significant difference in early reaction times between wearing half helmets and not wearing helmets (t = 0.86, $p = 0.391$). This indicates that compared to not wearing a helmet, wearing a full face helmet or 3/4 helmet shortens the early reaction time, while wearing a half helmet does not affect the early reaction time.

**Table 7.** Independent t-tests for early reaction times when wearing helmet or not (unit: seconds).

| Type | N | Mean | SD | t | P |
|---|---|---|---|---|---|
| Full face helmet | 359 | 0.96 | 0.485 | 10.60 | 0.000 |
| Not wearing a helmet | 120 | 1.58 | 0.578 |  |  |
| 3/4 helmet | 391 | 1.34 | 0.533 | 4.12 | 0.000 |
| Not wearing a helmet | 120 | 1.58 | 0.578 |  |  |
| Half helmet | 397 | 1.63 | 0.596 | −0.86 | 0.391 |
| Not wearing a helmet | 120 | 1.58 | 0.578 |  |  |

*3.7. Analysis of Number of Missed Presses with and without a Helmet*

Table 8 shows the results of the comparison of the number of missed presses with and without helmets. The number of missed presses was significantly greater when wearing a full face helmet compared to not wearing a helmet (t = 2.98, $p = 0.004$). However, there was no difference between wearing 3/4 helmets (t = 1.06, $p = 0.296$) or half helmets (t = 1.36, $p = 0.183$) and not wearing a helmet.

**Table 8.** Independent t-tests for missed presses when wearing helmet or not (unit: seconds).

| Type | N | Mean | SD | t | P |
|---|---|---|---|---|---|
| Full face helmet | 40 | 1.00 | 1.155 | 2.98 | 0.004 |
| Not wearing a helmet | 12 | 0.0 | 0.0 |  |  |
| 3/4 helmet | 40 | 0.23 | 0.733 | 1.06 | 0.296 |
| Not wearing a helmet | 12 | 0.0 | 0.0 |  |  |
| Half helmet | 40 | 0.08 | 0.350 | 1.36 | 0.183 |
| Not wearing a helmet | 12 | 0.0 | 0.0 |  |  |

## 4. Conclusions

The present study first investigated the early reaction times, the number of missed presses, and the number of false presses of participants when wearing different types of helmets. The three dependent variables when then analyzed with sex and age as independent variables. The conclusions are summarized as follows:

### 4.1. Early Reaction Time Experiment

1. Of the three types of helmet, wearing half helmets produced the longest early reaction times, followed by 3/4 helmets, with full face helmets with the shortest early reaction times. This indicates that there is a difference in reaction times when wearing different types of helmet.
2. The early reaction time when wearing a half helmet was the same as the early reaction time when not wearing a helmet. This indicates that reaction times are not affected when wearing a half helmet, but reaction times when wearing both a full face helmet and a 3/4 helmet were affected.
3. The results for the total number of missed and false presses when wearing the three types of helmet were the same as for the results of the early reaction time analysis. This also demonstrates the consistent results of all three experiments.
4. Neither sex nor age had an effect on reaction times in the present study. This may be due to the relatively small age distribution of the participants in the study.

### 4.2. Recommendations for Helmet Design

Half helmets only cover the top of the head. Because the face, chin, and back of the head are exposed, half helmets offer less protection. However, vision is not obstructed, and ventilation is also good. 3/4 helmets do not cover the chin, combining both safety and comfort. However, vision is slightly affected. Full helmets provide complete protection for the head. However, the field of vision and ventilation are poor, and the weight is also increased by the complete covering of the head.

As previously mentioned, according to Triggs and Harris [13], the 85th percentile distribution of driver braking time is between 1.26 and 3 s. The results of the present study show that the early reaction time for 3/4 helmets and half helmets (1.395 s and 1.622 s, respectively) are both be within this range, while the early reaction time (1.115 s) for full face helmets does not fall within this range. Therefore, we recommend that helmet design starts by increasing the percentage of the field of view.

There are many aspects of helmet design that need to be considered. The present study examined the effects of restricted vision and hearing. This can provide a reference to design helmets that at the same time as complying with regulatory requirements, striking a balance between strengthening the helmet and maximizing vision and hearing.

### 4.3. Recommendations for Future Research

1. Future experiments could increase the realism of the driving simulation, for example, through VR or actual riding on closed roads.
2. The length of the experiment can also be increased in the future to measure changes in reaction times or decreases in attention during long rides.
3. Future research can include different age groups to investigate the effect of age on riders' reaction times.

**Author Contributions:** Conceptualization, D.C.; Data curation, D.C.; Formal analysis, D.C.; Methodology, D.C.; Supervision, D.C.; Validation, D.C.; Writing—original draft preparation, Y.-H.C.; Writing—review and editing, C.-J.L. All authors have read and agreed to the published version of the manuscript.

**Funding:** This research received no external funding.

**Institutional Review Board Statement:** Not applicable.

**Informed Consent Statement:** Not applicable.

**Data Availability Statement:** Not applicable.

**Conflicts of Interest:** The authors declare no conflict of interest.

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
