# Peer review of "The Effects of Wearing Helmets on Reaction of Motorcycle Riders"

_vehicles, doi:10.3390/vehicles3040050_

Round 1

Reviewer 1 Report

No literature No 4

There are no references to the literature on motorcycle helmets, their durability, the scope of safety and legal provisions regarding this type of driver's head protection.

In the introduction, there is no extensive review of the literature, both in terms of traffic safety of cars and motorcycles or bicycles / two-wheeled vehicles /. There was no evidence of a lack of analyzes of motorcycle helmets in the literature. Helmet tests are subject to relevant regulations and must withstand the required range of impact forces. Manufacturers, in addition to endurance tests, design helmets in such a way as to ensure their correct putting on, keeping the driver on the head and assess the range of visibility. Additionally, an attempt can be made to assess which helmets are most often used in vehicles with low capacity and for small, urban, speed of movement, and which are used in high capacity motorcycles. This has consequences for the user of the bike.

2.2.1.

Mean response times misinterpreted. The table shows that Full face Helmet was the best. There are no descriptions of abbreviations used in the tables.

2.2.2

What is a "t-test"? no description of the statistical test and its boundary conditions. If it is stated that there are significant statistical differences between the results, how were they calculated and how much they are. The authors' thought lines cannot be traced. There are no references to the statistical test literature. The results presented in the tables show that the average value obtained is the best for the Full face helmet. Additionally, the average reaction time of 1.115 seconds is very good for the driver's reaction on the road. There is no literature related to the analysis of the driver's reaction time in road traffic, taking into account external disturbances and the psychophysical state of the driver. The entire chapter 2 requires a major overhaul. It lacks the continuity of the message, a precise evaluation of the obtained results and references to the literature. It would also be worth making tests in a full simulation of riding a motorcycle in controlled conditions.

  1.  

The boundary conditions are not accepted. They result from the scope of visibility of the human eye. Acute viewing angles and their ranges are given here in the literature. Then it will be possible to refer to the range of visibility in motorcycle helmets. The comparative test / GoPro camera / is correct, but without reference to the possibility of the eye, it is illustrative. Failure to define the extent of the blind spot. It should also be indicated when the given range of visibility is sufficient for safe movement on the road. Is full visibility necessary / driving without a helmet /? Do the indicated visibility limitations have a significant impact on the perception of random events on the road? No such relationship is visible from the average values ​​of the reaction time (the open helmet was the worst). An important comparative drawing would be the correlation of the ranges of visibility of the organ of vision without a helmet and in the 3 types of helmets. It would also allow to estimate the distance from the driver to the obstacle to be noticed. This is of particular importance in road safety and, in the event of an accident, for prosecutors and courts.

The field of view in the half of the helmet and the helmet ¾ is almost identical, 200x300cm and 200x305cm. In contrast, the field of view proportion differs by 28%. Why is there such a big difference? One gets the impression that the values ​​expressed as a percentage of the field of view do not correspond to the obtained test results.

Table 15 shows that the best response time, despite the smallest field of view range, is obtained when riding in Full face helmet / 1,115s area 50% /. This proves that such a helmet is the best and, additionally, the safest for the rider.

  1.  

The results were described, but the direction of further actions was not indicated.

There is no reference to the theses contained in the abstract: The experimental results can be used as a reference for helmet design and academic research. There are no references to the literature determining the location of research in both scientific and practical and industrial obsolescence.

Literature

Autocit 0/3

2/3 from before 2011.

1/3 from before 2000.

There is no broad spectrum of current literature on the subject.

Authors should try to ensure that the literature older than 10 years does not exceed 25%.

Author Response

Response to Reviewer 1 Comments

Point 1:

There are no references to the literature on motorcycle helmets, their durability, the scope of safety and legal provisions regarding this type of driver's head protection.

In the introduction, there is no extensive review of the literature, both in terms of traffic safety of cars and motorcycles or bicycles / two-wheeled vehicles /. There was no evidence of a lack of analyzes of motorcycle helmets in the literature. Helmet tests are subject to relevant regulations and must withstand the required range of impact forces. Manufacturers, in addition to endurance tests, design helmets in such a way as to ensure their correct putting on, keeping the driver on the head and assess the range of visibility. Additionally, an attempt can be made to assess which helmets are most often used in vehicles with low capacity and for small, urban, speed of movement, and which are used in high capacity motorcycles. This has consequences for the user of the bike.

Response 1:

From line 12-14 of paragraph 1 on page 1 to line 1-14 on page 2. The text is as follows:

After Taiwan enforced the mandatory use of helmets for motorcycle riders and passengers on June 1, 1997 [3], the helmet usage rate has reached 80%. In addition, the criteria for approved helmets and the manner in which helmets are to be worn are also clearly stipulated [4]. As a result, helmets have become an essential safety item when riding motorcycles in Taiwan.

In Taiwan, helmets must pass the National Standards of the Republic of China (CNS) and the Product Safety Mark [5], which are divided into two types of helmet: normal (suitable for non-racing motorcycles under 125cc) and strengthened (suitable for non-racing motorcycles over 125cc). In addition, helmets approved by the United States Department of Transportation (DOT) [6], Economic Commission of Europe (ECE) [7], SNELL Foundation 2020 Helmet Standard For Use in Motorcycling (SNELL M2020) [8], and Japanese Industrial Standards (JIS) [9] can also be sold. The DOT adopts independent certification and conducts random testing after the market launch, and is the most common helmet certification in the world. SNELL’s certification process is more stringent than DOT, and helmets that pass the DOT standards may not pass SNELL testing. In addition, SNELL purchases helmets on the market with SNELL labels for random retesting [10].

Point 2:

Mean response times misinterpreted. The table shows that Full face Helmet was the best. There are no descriptions of abbreviations used in the tables.

Response 2:

Page 2, paragraph 4, page 5, section 3.1, paragraph 1, and section 3.1, paragraph 4. The text is as follows:

Robers [11] described the sum of the time for a driver to perceive, assess, judge, and react to road conditions as the response time. This response time is approximately 2–4 seconds. Nicholas and Lester [12] found that reaction times are an important factor in determining braking distance and are related to the stopping sight distance. When the reaction time is too slow, the driver is unable to brake in time, causing an accident. In other words, when drivers react more quickly, they have more time to make an appropriate judgment (such as braking or taking evasive action).

The early reaction times for the three types of helmets are shown in Table 1. Half helmets had the longest early reaction time, with the object detected on average 1.622 seconds in advance. 3/4 helmets had the next longest early reaction time, with the object detected on average 1.395 seconds in advance. Full helmets had the shortest early reaction time, with the object detected on average 1.115 seconds in advance.

Of the three types of helmet, full face helmets had the shortest early reaction time, 3/4 helmets had the next shortest early reaction time, and half helmets had the longest early reaction time. Therefore, wearing full face helmets shortens riders’ early reaction time in case of an incident.

Point 3:

What is a "t-test"? no description of the statistical test and its boundary conditions. If it is stated that there are significant statistical differences between the results, how were they calculated and how much they are. The authors' thought lines cannot be traced. There are no references to the statistical test literature. The results presented in the tables show that the average value obtained is the best for the Full face helmet. Additionally, the average reaction time of 1.115 seconds is very good for the driver's reaction on the road. There is no literature related to the analysis of the driver's reaction time in road traffic, taking into account external disturbances and the psychophysical state of the driver. The entire chapter 2 requires a major overhaul. It lacks the continuity of the message, a precise evaluation of the obtained results and references to the literature. It would also be worth making tests in a full simulation of riding a motorcycle in controlled conditions.

Response 3:

Page 5, section 2.4, paragraph 4, section 3.1, paragraph 2, section 3.1, paragraph 4, and page 2, section 2.1, paragraph 1. The text is as follows:

The analysis of the two experiments was conducted by a statistical analysis t-test. The t-test was proposed by William Sealy Gosset (pseudonym “Student”) in 1908 [13].  When conducting research, the t-test is the most commonly used statistical method to determine whether there is a significant difference between two sets of data [14]. The experimental group and control group can be considered as two independent samples for testing the hypothesis using the sample means (two independent samples test) [15].

According to Triggs and Harris [16], the 85th percentile distribution of driver braking time is between 1.26 and 3 seconds. In the present study, only the early reaction time of half helmets was within the 85th percentile distribution of braking time as measured by Triggs and Harris.

Paired t-tests were used to test the early reaction times of the three types of helmet. The test results are shown in Table 2. The early reaction time wearing full face helmets was shorter than when wearing 3/4 helmets (t= 21.54, p< 0.01) and half helmets (t= 28.38, p< 0.01). The early reaction time when wearing 3/4 helmets was shorter than when wearing half helmets (t = 18.54, p< 0.01).

The present study recruited 52 adult participants, with 26 participants of each sex. The age of participants was between 18–35 years old, with an average age of 23.24 years old (SD= 3.54 years old). All participants had motorcycle licenses and had 1.0 vision of better after correction. The participants had good mental and physical condition at the time of the experiment and did not stay up late or overwork prior to participating in the experiment.

Chapter 2 and Chapter 3 have been revised and reorganized, please reviewer for review.

Point 4:

The boundary conditions are not accepted. They result from the scope of visibility of the human eye. Acute viewing angles and their ranges are given here in the literature. Then it will be possible to refer to the range of visibility in motorcycle helmets. The comparative test / GoPro camera / is correct, but without reference to the possibility of the eye, it is illustrative. Failure to define the extent of the blind spot. It should also be indicated when the given range of visibility is sufficient for safe movement on the road. Is full visibility necessary / driving without a helmet /? Do the indicated visibility limitations have a significant impact on the perception of random events on the road? No such relationship is visible from the average values ​​of the reaction time (the open helmet was the worst). An important comparative drawing would be the correlation of the ranges of visibility of the organ of vision without a helmet and in the 3 types of helmets. It would also allow to estimate the distance from the driver to the obstacle to be noticed. This is of particular importance in road safety and, in the event of an accident, for prosecutors and courts.

Response 4:

All deleted.

Point 5:

The results were described, but the direction of further actions was not indicated.

There is no reference to the theses contained in the abstract: The experimental results can be used as a reference for helmet design and academic research. There are no references to the literature determining the location of research in both scientific and practical and industrial obsolescence.

Response 5:

Page 9, Section 4.2, Paragraph 1-3. The text is as follows:

Half helmets only cover the top of the head. Because the face, chin, and back of the head are exposed, half helmets offer less protection. However, vision is not obstructed, and ventilation is also good. 3/4 helmets do not cover the chin, combining both safety and comfort. However, vision is slightly affected. Full helmets provide complete protection for the head. However, the field of vision and ventilation are poor, and the weight is also increased by the complete covering of the head.

As previously mentioned, according to Triggs and Harris [13], the 85th percentile distribution of driver braking time is between 1.26 and 3 seconds. The results of the present study show that the early reaction time for 3/4 helmets and half helmets  (1.395 seconds and 1.622 seconds, respectively) are both be within this range, while the early reaction time (1.115 second) for full face helmets does not fall within this range. Therefore, we recommend that helmet design starts by increasing the percentage of the field of view.

There are many aspects of helmet design that need to be considered. The present study examined the effects of restricted vision and hearing. This can provide a reference to design helmets that at the same time as complying with regulatory requirements, striking a balance between strengthening the helmet and maximizing vision and hearing.

The literature has been re-edited.

Point 6:

Literature

Autocit 0/3

2/3 from before 2011.

1/3 from before 2000.

There is no broad spectrum of current literature on the subject.

Authors should try to ensure that the literature older than 10 years does not exceed 25%.

Response 6:

The literature has been re-edited, please review pages 9-10.

Reviewer 2 Report

The topic is very actual and important for the understanding of the effects of helmets on reaction times of motorcycle riders as well as one of the major road safety problems. Based on the submitted paper, I am sending you my essential comments. 

The comments are below:

  • The heading "Introduction" is very poor- a quality literature review is lacking; The Reference list is too poor.
  • Please explain Fig. 2- Scenario A and Scenario B;
  • The heading “Reaction Time Experiment” is not concise. Please, shorten that heading (too many subheadings and Tables).
  • The results need to be better presented and explained as well as better visualized. Try to answer why such results were obtained.
  • Please, improve the heading “Conclusion”. Under that heading, explain the results, any limitations and future research.

After the improvement of the paper in line with all comments, I suggest publishing the paper.

Author Response

Response to Reviewer 2 Comments

Point 1:

The heading "Introduction" is very poor- a quality literature review is lacking; The Reference list is too poor.

Response 1:

Introduction has been revised, please review pages 1-2. The text is as follows:

According to the National Travel Survey carried out by the Ministry of Transportation and Communications [1], in 2016, the most commonly used mode of transportation in Taiwan was motorcycles (45.9%). According to data released by the National Police Agency [2], there were 238,780 deaths and injuries involving motorcycle riders in 2014, accounting for 57.53% of traffic accident deaths and injuries. In addition, the percentage of traffic accident deaths and injuries involving motorcycle rides exceeded 50% for three consecutive years starting in 2012. Based on the above data, motorcycles are the most commonly used means of transportation and also have the highest accident rate. Although motorcycles have the advantages of being economical and convenient, because motorcycles are less stable and provide less protection than cars, motorcycle riders are vulnerable in traffic accidents. Furthermore, head trauma is often fatal, causing great loss to society. After Taiwan enforced the mandatory use of helmets for motorcycle riders and passengers on June 1, 1997 [3], the helmet usage rate has reached 80%. In addition, the criteria for approved helmets and the manner in which helmets are to be worn are also clearly stipulated [4]. As a result, helmets have become an essential safety item when riding motorcycles in Taiwan.

In Taiwan, helmets must pass the National Standards of the Republic of China (CNS) and the Product Safety Mark [5], which are divided into two types of helmet: normal (suitable for non-racing motorcycles under 125cc) and strengthened (suitable for non-racing motorcycles over 125cc). In addition, helmets approved by the United States Department of Transportation (DOT) [6], Economic Commission of Europe (ECE) [7], SNELL Foundation 2020 Helmet Standard For Use in Motorcycling (SNELL M2020) [8], and Japanese Industrial Standards (JIS) [9] can also be sold. The DOT adopts independent certification and conducts random testing after the market launch, and is the most common helmet certification in the world. SNELL’s certification process is more stringent than DOT, and helmets that pass the DOT standards may not pass SNELL testing. In addition, SNELL purchases helmets on the market with SNELL labels for random retesting [10].

Although helmets provide protection to the head, they also affect the visual field of motorcycle riders. Currently, helmets available on the market can be divided into half helmets, 3/4 helmets, and full face helmets. Different types of helmets will produce different obstructions to the visual field of riders, and this visual field is closely related to riders’ reaction times.

Robers [11] described the sum of the time for a driver to perceive, assess, judge, and react to road conditions as the response time. This response time is approximately 2–4 seconds. Nicholas and Lester [12] found that reaction times are an important factor in determining braking distance and are related to the stopping sight distance. When the reaction time is too slow, the driver is unable to brake in time, causing an accident. In other words, when drivers react more quickly, they have more time to make an appropriate judgment (such as braking or taking evasive action).

However, the literature mostly focuses on the protective effect of helmets after a collision and rarely considers the influence of helmets prior to collisions. Therefore, the present study examined riders’ early reaction times, false presses, missed presses, sex, and age for three types of helmets, analyzing possible influence factors as a starting point for improving helmet design.

Point 2:

The heading "Introduction" is very poor- a quality literature review is lacking; The Reference list is too poor.

Response 2:

Page 2, paragraph 4, page 4 and page 6, paragraph 2. The text is as follows:

Robers [11] described the sum of the time for a driver to perceive, assess, judge, and react to road conditions as the response time. This response time is approximately 2–4 seconds. Nicholas and Lester [12] found that reaction times are an important factor in determining braking distance and are related to the stopping sight distance. When the reaction time is too slow, the driver is unable to brake in time, causing an accident. In other words, when drivers react more quickly, they have more time to make an appropriate judgment (such as braking or taking evasive action).

â‘  Participants’ visual acuity was measured before the experiment. Participants with corrected visual acuity of better than 1.0 were allowed to perform the experiment. â‘¡ Participants sat in front of a table 150 cm away from the screen. The height of the chair was adjusted so that participants’ eyes were at the same height as the center of the screen. Participants were informed that the experiment was a video viewing experiment to simulate reactions when riding a motorcycle. Warning triangles appeared on both sides of the road in the video. Participants were asked to press a button when they saw a warning triangle to confirm that they had seen the warning triangle (as shown in Figure 4). â‘¢ A computer program was used to control the playback of the video. When participants pressed the button, the time and the location of the press were recorded. â‘£ At the end of the experiment, the early reaction time (as shown in Figure 5), number of false presses, and number of missed presses of each participant were recorded.

Of the three types of helmet, full face helmets had the shortest early reaction time, 3/4 helmets had the next shortest early reaction time, and half helmets had the longest early reaction time. Therefore, wearing full face helmets

Point 3:

The heading “Reaction Time Experiment” is not concise. Please, shorten that heading (too many subheadings and Tables).

Response 3:

This part has been revised and reorganized, please review 3. Results and Discussions  (pages 5-8).

Point 4:

The results need to be better presented and explained as well as better visualized. Try to answer why such results were obtained.

Please, improve the heading “Conclusion”. Under that heading, explain the results, any limitations and future research.

Response 4:

This part has been revised and reorganized, please review 4. Conclusions (pages 8-9).

Round 2

Reviewer 1 Report

The article has been heavily modified. As a result, it was necessary to evaluate it from the beginning. The title itself has already changed, which has made its message more specific.

Two-wheelers run in the so-called unbalanced range. The stability of the movement of, for example, a motorcycle depends, among other things, on the speed, structure and type of surface or the type and condition of tires. According to the authors' suggestion, the driver's reaction time should be added to the scope and quantification of safety features. The authors correctly noticed that the driver's reaction time depends, among other things, on the visibility of the surroundings. However, there is no reference to the literature specifying the driver's reaction time depending on the psycho-motor condition of the driver, the condition and type of road surface and the environment, as well as the conditions of traffic intensity or the impact of weather. Such literature is indeed mainly related to driving a car, however, it is an important reference point and the possibility of comparing the obtained reaction times. Such tests are also carried out on driving simulators, where the driver's attention is specially distracted and various road situations are simulated. This allows for the development of driver assistance systems and has an impact on road infrastructure regulations.

It would also be important to refer to the research literature related to motorcycle driver assistance systems and security that can be used by the driver himself / special clothing, footwear, gloves or designed airbags for motorcyclists /. One can also refer to the research on motorcycle acceleration in terms of traffic safety. Such a wide spectrum should be focused on the presented research on reaction times and its impact on the safety of two-wheeled traffic. The methodology proposed in the article has been thought out by the authors and is clear and understandable. It is an interesting and practical approach to assessing the driver's reaction in pseudo-real conditions. Here, attention should be paid to the comfort of the test participant who did not have to drive a two-wheeler, which was done for safety reasons. Relatively young participants were selected for the study. It is understandable that most of the drivers are of this age group, but it would be imperative to perform such a test also on a group of elderly people. It would also provide valuable information on whether the driver's reaction time changes with the age of the driver and whether this change has a significant impact on the scope of traffic safety.

Author Response

Point 1:

Two-wheelers run in the so-called unbalanced range. The stability of the movement of, for example, a motorcycle depends, among other things, on the speed, structure and type of surface or the type and condition of tires. According to the authors' suggestion, the driver's reaction time should be added to the scope and quantification of safety features. The authors correctly noticed that the driver's reaction time depends, among other things, on the visibility of the surroundings. However, there is no reference to the literature specifying the driver's reaction time depending on the psycho-motor condition of the driver, the condition and type of road surface and the environment, as well as the conditions of traffic intensity or the impact of weather. Such literature is indeed mainly related to driving a car, however, it is an important reference point and the possibility of comparing the obtained reaction times. Such tests are also carried out on driving simulators, where the driver's attention is specially distracted and various road situations are simulated. This allows for the development of driver assistance systems and has an impact on road infrastructure regulations.

Response 1:

From paragraph 5, 6, 8 on page 2 and paragraph 1, 2 on page 3. The text is as follows:

Other factors affecting riders’ early reaction times include the rider’s mental state and age, road pavement, traffic intensity, and weather conditions.

Research on driving fatigue generally argues that the length of time spent driving is the most relevant factor. The fatigue effect begins to occur when driving continuously for more than eight hours.[13][14] In addition, according to medical data, the body begins to experience physical fatigue after one hour of monotonous and repetitive driving regardless of whether the individual is aware of it.[15] In order to avoid the occurrence of these factors that may affect the results of the experiment, we ensured that participants were in good mental and physical condition at the time of the experiment, and the total length of the video was limited to ten minutes

Currently, most road pavements in Taiwan use asphalt concrete pavement and portland cement concrete pavement. The former is constructed in accordance with the “Standard Specifications for Highway Construction Section 02742”[19] and AASHTO.[20] The latter is constructed in accordance with the “Standard Specifications for Highway Construction Section 02751”[19] and ASTM.[21] Therefore, road pavements have fatigue resistance (the ability to prevent the pavement breaking up due to the bending effect of repeated vehicle loads) and skid resistance (resistance to sliding when the brakes are applied), reducing the risk of riders sliding and providing safe road performance for road users.

Past studies have shown that the accident frequency of each road segment is mainly influenced by factors related to annual average daily traffic, road geometry, and weather conditions.[22][23]24][25] However, there is strong evidence showing that human factors are the most important influencing factor in road traffic accidents.[26][27][28][29][30][31] Distraction and inattention are the two most important human factors in road traffic accidents.

Based on the above, the participants in this study were young riders who have a higher ratio of accidents. Watching a pre-recorded video excluded factors such as road pavement, traffic intensity, and weather conditions, allowing us to focus on the field of vision and early reaction time when wearing helmets.

Point 2:

It would also be important to refer to the research literature related to motorcycle driver assistance systems and security that can be used by the driver himself / special clothing, footwear, gloves or designed airbags for motorcyclists /. One can also refer to the research on motorcycle acceleration in terms of traffic safety. Such a wide spectrum should be focused on the presented research on reaction times and its impact on the safety of two-wheeled traffic. The methodology proposed in the article has been thought out by the authors and is clear and understandable. It is an interesting and practical approach to assessing the driver's reaction in pseudo-real conditions. Here, attention should be paid to the comfort of the test participant who did not have to drive a two-wheeler, which was done for safety reasons. Relatively young participants were selected for the study. It is understandable that most of the drivers are of this age group, but it would be imperative to perform such a test also on a group of elderly people. It would also provide valuable information on whether the driver's reaction time changes with the age of the driver and whether this change has a significant impact on the scope of traffic safety.

Response 2:

Page 2, paragraph 7, and page10, paragraph 3. The text is as follows:

Age is considered to be one of the influencing factors in previous studies on the physiological condition of drivers. Older drivers are less able to perceive potential road hazards due to deteriorating physiological function (e.g., concentration, effective visual field, physical strength, and reflexes).[16] However, Borowsky et al. [17] analyzed the effect of age and driving experience on the ability to detect hazards, finding that older and more experienced drivers are more sensitive to hazardous situations than younger drivers. In addition, statistics have shown that older drivers (aged 60 and over) account for 15.9% of fatalities and injuries, compared to 43.2% involving younger drivers (aged 18–29).[18]

Future research can include different age groups to investigate the effect of age on riders’ reaction times.

Round 3

Reviewer 1 Report

The authors largely answered the questions posed. I believe that a literature-based article prepared in such a way correctly shows the issue raised.